# Biocompatible Iron Oxide Nanoparticles for Targeted Cancer Gene Therapy: A Review

**DOI:** 10.3390/nano12193323

**Published:** 2022-09-24

**Authors:** Jinsong Zhang, Tianyuan Zhang, Jianqing Gao

**Affiliations:** 1College of Pharmaceutical Sciences, Zhejiang University, Hangzhou 310058, China; 2Department of Pharmacy, the Second Affiliated Hospital, School of Medicine, Zhejiang University, Hangzhou 310058, China

**Keywords:** iron oxide nanoparticles, gene delivery, tumor targeting, cancer treatment, tumor diagnosis

## Abstract

In recent years, gene therapy has made remarkable achievements in tumor treatment. In a successfully cancer gene therapy, a smart gene delivery system is necessary for both protecting the therapeutic genes in circulation and enabling high gene expression in tumor sites. Magnetic iron oxide nanoparticles (IONPs) have demonstrated their bright promise for highly efficient gene delivery target to tumor tissues, partly due to their good biocompatibility, magnetic responsiveness, and extensive functional surface modification. In this review, the latest progress in targeting cancer gene therapy is introduced, and the unique properties of IONPs contributing to the efficient delivery of therapeutic genes are summarized with detailed examples. Furthermore, the diagnosis potentials and synergistic tumor treatment capacity of IONPs are highlighted. In addition, aiming at potential risks during the gene delivery process, several strategies to improve the efficiency or reduce the potential risks of using IONPs for cancer gene therapy are introduced and addressed. The strategies and applications summarized in this review provide a general understanding for the potential applications of IONPs in cancer gene therapy.

## 1. Introduction

Cancer is one of the most significant global threats to human life and health. Surgery, chemotherapy, and radiation therapy are the mainstays of clinical tumor treatment. However, the complexity of tumors and their microenvironment imposes limitations on the therapeutic efficiency using conventional treatment strategies. In addition, chemotherapy and radiation therapy may kill both tumor cells and normal cells due to their lack of specificity, resulting in several adverse side effects [1,2]. As a novel cancer treatment method, gene therapy can effectively inhibit tumor proliferation and growth by directly regulating key genes or proteins (e.g., KRAS, cMYC, MDR1, and PTEN), demonstrating attractive therapeutic effects in treating a variety of cancers, including lung, liver, breast, brain, and gastrointestinal tumors [2,3,4]. Notably, tumor suppression can be achieved in some gene therapy cases by modulating the tumor cell microenvironment by targeting tumor−associated angiogenesis, fibroblasts, and immune responses [5].

DNAs, small interfering RNAs (siRNAs), microRNAs (miRNAs), and messenger RNAs (mRNAs) are the most frequently used nucleic acid drugs for gene therapy [6,7,8,9,10]. In the cytoplasm, siRNA−induced silencing complexes are capable of particular complementary binding to target mRNA selectively and subsequent silencing of cancer−related genes. miRNAs can bind to particular mRNAs to inhibit their translation or promote degradation [11]. Unlike other pathways, DNA mainly enters the nucleus, where it regulates the expressions of target proteins through further transcription and translation [12]. Moreover, gene−editing technology provides more opportunities for cancer gene therapy. Using CRISPR/Cas9, an efficient gene−editing system consisting of single−guide RNA and Cas9 protein, target genes in tumors can be deleted, inserted, or modified to produce significant antitumor therapeutic effects [13,14].

Safe, effective, and programmable vectors are crucial to the success of gene therapy [15,16]. Direct delivery of naked gene drugs is susceptible to attack by various nucleases, resulting in less accumulation of gene drugs at the tumor sites [17]. Because of their considerable molecular weight, negative charges, and hydrophilicity, naked genes also have difficulty in entering cancer cells [18,19]. In vivo, naked genes are easily recognized and destroyed by extracellular and intracellular endonucleases, exonucleases, and the innate immune system [20,21,22]. Meanwhile, due to the inability of tumor-targeting, naked gene drugs may affect normal cells and tissues, thereby posing safety risks [23,24]. Consequently, selecting an appropriate gene delivery vehicle is necessary for successful cancer gene therapy.

## 2. IONPs for Gene Delivery Target to Cancer Cells

Currently, viral vectors, such as adenovirus, retrovirus, vaccinia virus, and herpes virus, are the most applied vectors for gene delivery, and more than 60% of clinical treatment trials employ viral vectors [18,25]. Viral vectors have been used to deliver therapeutic genes to cancer cells, and four viral−vector−based gene therapies have been clinically approved, including Talimogene laherparepvec, Mx−dnG1, H101, and Ad−p53 [26]. An important challenge in the current clinical trials of viral−vector−based gene therapy is the safety hazard from the innate or adaptive immune response caused by viral vectors. In addition, high production costs may be encountered when using viral vectors for large−scale production, limiting the clinical translation for gene delivery [27,28,29,30].

Using non−viral agents, such as polymers, liposomes, and micelles, to facilitate gene entry into tumor cells has garnered considerable attention [31,32]. Liposomes and nanoparticles are the most extensively researched non−viral carriers, showing major advantages to improve gene delivery efficiency, but are facing challenges of their short half−life, low biological activity, and potential hepatotoxicity [33,34]. In addition to liposomes, polyplex micelles encapsulating genetic drugs have been actively studied in cancer therapy, using anti−angiogenic [35,36,37] and suicide gene delivery [38,39,40]. However, limited blood circulation half−life due to liver sinusoidal capture remains the major critical hurdle [41,42]. Inorganic nanomaterials, such as gold, silver, and silicon dioxide, are recently developed for efficient gene delivery, showing advantages of controllable size and good uniformity [43]. However, it should be noted that these inorganic nanocarriers tend to accumulate in the liver [44] and kidney, resulting in potential toxicity, which limits their use as gene delivery vehicles [45,46,47].

Magnetic IONPs have more potential for tumor−targeted gene delivery than other inorganic nanomaterials due to their exceptional properties. These properties include superior magnetic responsiveness, simple preparation, ease of chemical functionalization, low toxicity, and high biocompatibility [48,49,50,51]. Luo et al. applied amino−ester lipid−modified superparamagnetic iron oxide nanoparticles (SPIONs) to deliver mRNA, showing advantages in a high mRNA−mediated protein expression and an increased *r2* relaxivity for magnetic resonance imaging (MRI). Hence, they thought IONPs had promising potential as delivery vehicles of mRNA for theranostic applications [9]. Research from Wang et al. demonstrated that IONPs coated with chitosan−polyethylene (PEG) had the ability to deliver siRNA to hepatocellular carcinoma in vitro and suppressed Luc expression. In a xenograft mouse model, the nanovector could specifically bind to tumors and induce remarkable inhibition of Luc expression [52].

Through electrostatic adsorption, surface−modificated IONPs are loaded with large quantities of negative−charged genetic drugs. The therapeutic genes carried by IONPs can then be delivered to tumor cells under the navigation of an external magnetic field (Figure 1). IONPs have demonstrated their promising ability for targeting gene delivery to different types of tumors, such as breast cancer, glioma, cervical cancer, prostate cancer, and gastric cancer (Table 1). Thus far, lots of studies have indicated the potent promise of IONPs in the treatment of solid tumors. 

For example, Zhang et al. recently showed that IONPs could deliver siRNA to intracranial xenografted glioma and effectively reduced glutathione peroxidase 4 expression, resulting in a significantly improved inhibition of tumor growth [62]. Yang et al. found that using galactose− and polyetherimide−modified IONPs could largely enhance siRNA accumulation in orthotopic solid tumors for as long as 24 h after intravenous injection, resulting in a significant reduction in the volume of hepatocellular carcinoma, as well as liver/body weight ratio [63]. Moreover, Dan et al. recently demonstrated the promising potential of galactose (Gal)−and−polyethyleneimine (PEI)−coated SPIONs (Gal−PEI−SPIONs) for gene delivery and MRI in an in situ hepatic tumor model. The Gal−PEI−SPIONs showed the ability to selectively deliver siRNA−encoding telomerase reverse transcriptase genes to tumor sites in the liver after systemic injection, thereby showing a significant inhibition of tumor growth [64]. In addition, IONPs have also demonstrated the ability to inhibit tumor metastasis in animal models [65,66,67]. For example, after intravenous injection in the early stage of metastasis development, IONPs modified with the tumor−penetrating peptide iRGD could target and reduce tumor growth in the brain [66]. Zhang et al. applied IONPs modified with 3−aminopropyltriethoxysilane for tumor−targeted delivery of cytosine−phosphate−guanine (CpG), a novel toll−like receptor 9 (TLR9) agonist, showing a good therapeutic effect to inhibit metastasis in 4T1 breast cancer [67].

The enhanced permeability and retention (EPR) effect is the ability of nanoparticles between 10 and 100 nm to passively penetrate the tumor site’s complex and dense microvascular structure and persist in the tumor tissue for an extended period without being cleared by the lymphatic system [68]. The size of the nanoparticles plays an important role in the EPR effect of the nanocarriers [69]. Advantages of the controllable size of IONPs enable a better utilization of the EPR effect can be realized to deliver genetic drugs carrying nanoparticles targeting tumor sites. Different from the nonspecific mechanisms based on the EPR effect, the active targeting mechanisms approaching peptides, antibodies, and small molecules also play an important role in tumor targeting. Due to the exceptional surface functionalization capacity of IONPs, modified IONPs can bind to specific receptors on tumor cells, thereby facilitating the active targeting of IONP gene complexes to tumor sites [55,70,71]. In addition, the magnetic targeting of IONPs using gradient magnetic field gradient is crucial to drive IONPs actively target to the tumor tissue [72,73]. The properties of the magnetic field closely relate to the efficiency of magnetic targeting [74]. An external and static magnetic field between 0.2 and 0.6 T was reported to guide IONPs toward the tumor region [75]. The strength and location of the external magnetic field influence the magnetic response of IONPs, further affecting the efficiency of tumor−targeted gene delivery [50,65]. Solid tumors with a certain shape and position may facilitate the setting of an external magnetic field, thereby improving the magnetic targeting of nanoparticles to tumor sites and the accumulation of genetic drugs.

## 3. IONPs for Tumor Diagnosis and Combination Therapy

It has been reported that IONPs have functional advantages in cancer treatment, including magnetic resonance imaging, photothermal, photodynamic, magnetocaloric, and immune activation [76,77,78]. Combinations of gene therapy with imaging, chemotherapy drug delivery, enhanced immune response, radiotherapy, and phototherapy based on IONPs can be used to diagnose and treat tumors in a synergistic treatment (Figure 2).

### 3.1. IONPs for Tumor Cell Selected Imaging

By influencing transverse relaxation (T2), magnetic IONPs with a size of less than 30 nm, are capable of being controlled by electromagnetic fields (EMFs) and used as contrast agents in MRI systems [79,80,81]. For example, due to the coupled spins of 3 d electrons unpaired with the cubic lattice of Fe^3+^ and Fe^4+^ cations, SPIONs can flip the orientation of their core protons when exposed to electromagnetic fields. This results in local field inhomogeneities and negative contrast in T2−weighted imaging, which enables tissue imaging with high contrast and spatial resolution in MRI systems [82].

The imaging capabilities of IONPs confer significant advantages over other gene delivery vehicles, such as providing accurate diagnostic information for tumor lesion localization, which is crucial for precise and targeted therapy. MRI technology allows the observation of the biological distribution and pharmacokinetic properties of IONPs carrying gene drugs in various organ systems, indicating the gene delivery efficiency to the tumor site [83,84,85]. Mahajan et al. designed a siPLK1−conjugated streptavidin−conjugated dextran−coated SPION (siPLK1−StAv−SPION) delivery platform through directly knocking down cell−cycle−specific serine−threonine−kinase. This platform inhibited tumor cell apoptosis and proliferation by tumor−specific silencing of PLK1 and allowed for the non−invasive assessment of in vivo delivery efficiency by imaging tumor response (Figure 3) [86].

### 3.2. IONPs for Co−Delivery of Therapeutic Genes and Chemotherapeutic Drugs

Chemotherapy drugs widely used in clinics today, such as cisplatin, paclitaxel, and doxorubicin (DOX), have a good killing effect on cancer cells but can also damage normal cells and tissues and cause noticeable side effects [87]. In recent years, the strategy of utilizing IONPs for the combined delivery of gene drugs and chemotherapeutic drugs has been demonstrated to be both feasible and highly effective [88,89,90,91]. Li et al. used PEG−PEI−coated IONPs to deliver microRNA−21 antisense oligonucleotide (ASO−miR−21) and gemcitabine (Gem) to pancreatic cancer cells. They observed that this co−delivery strategy effectively inhibited the growth and metastasis of tumor cells via the upregulation of tumor suppressor genes PDCD4 and PTEN and the inhibition of epithelial–mesenchymal transition [92]. Co−loading gene and chemical drugs on surface−modified IONPs not only enhances the killing effect on tumor cells via various mechanisms and pathways but also prevents damage from drugs to normal cells or tissues.

### 3.3. IONPs for Inducing Antitumor Immune Response

The immune response can be often triggered after exogenous gene−drug−carrying IONPs are introduced into the body. Recent studies have shown that this immune activation induces the synergistic immune response to recognize and kill tumors, although this immune response is reduced in most drug delivery vehicle designs [93,94]. When exogenous genes are endocytosed, toll−like receptors in the immune system are activated, recognizing the genetic drugs and activating the type I interferon signaling pathway, which can activate various immune responses. Immune cells such as macrophages, T lymphocytes, B lymphocytes, and dendritic cells can be combined with chemotherapy drugs to kill tumor cells [95,96].

In a recent study, Meng et al. co−delivered peptide antigens and adjuvants (cytosine 5′ to a guanine dinucleotide repeat; CpG DNA) to dendritic cells (DCs) in the cytosol and lysosomes via membrane fusion and endocytosis with lipid−coated iron oxide nanoparticles (IONP−C/O@LPs), synergistically activating immature DCs. In addition, IONPs appeared to promote DC maturation by generating intracellular reactive oxygen species during this process. Through subcutaneous injection, IONP−C/O@LPs accumulated in draining lymph nodes activated immature DCs efficiently and increased antigen−specific T cells in tumors and the spleen, inducing local and systemic antitumor immune responses and inhibiting tumor growth (Figure 4) [97].

### 3.4. IONPs for Combined Phototherapy

Phototherapy, including photothermal therapy treatment (PTT) and photodynamic therapy treatment (PDT), is emerging as a promising strategy for the repeatable and accurate ablation of tumor cells. As potential photothermal materials, IONPs can convert radiated energy into heat to raise the temperature above 42 °C, resulting in the effective killing of tumor cells [98,99]. IONPs also have demonstrated their capabilities to improve the efficiency and safety of this therapeutic strategy by targeting the delivery of photosensitizing agents to tumor sites in animal experiments. For examples, Li et al. demonstrated that IONPs modified with the potent photosensitizer Chlorin e6 could significantly increase the distribution and retention of photosensitizers in mouse subcutaneous melanoma grafts and enhance photodynamic therapy effect [100]. Further, resent research from Yu et al. reported Chlorin e6 loaded through functionalized IONPs linked with glucose showed both target photodynamic efficacy and enhancement in immunogenicity in lung cancer [101]. In addition, IONPs were reported to enhance treatment efficacy in head and neck xenograft tumors and, more importantly, to reduce photosensitizer dose to avoid PDT potential toxicity to normal tissues [102]. However, the advantages of IONPs in targeted phototherapy against tumors have so far only been observed in experimental models, and further verifications of the effectiveness of IONPs in clinical settings are necessary.

IONPs have the potential to combine gene therapy and phototherapy to achieve a synergistic tumor−killing effect. Huang et al. applied IONPs as a bridge to combine PTT and gene therapy. They loaded porous iron oxide nanoparticles (PIONs) with pcDNA3.1−vector−encoding long noncoding RNA crystallin beta−gamma domain−containing 3 (LNC CRYBG3) to overexpress LNC CRYBG3 in tumor cells for degrading the actin cytoskeleton, and inducing cell apoptosis, resulting in the effective destruction of non−small−cell lung cancer cells in vitro and in vivo (Figure 5) [103]. In addition, the success of some IONP−based chemotherapeutics and phototherapy synergistic therapies in animal tumor models also provide the promising of the combination of gene therapy and phototherapy on the basis of IONPs [104,105]. However, it is essential to note that, due to the limited permeability of light to tissues, infrared−light−based PDT and PTT are ineffective in treating deep−seated tumors.

### 3.5. IONPs for Combined Radiation Therapy

Tumor−targeting radiation therapy is one of the most frequently employed cancer therapies in clinical practice [106]. Definitive radiation therapy consists of high−energy rays that only deliver a lethal dose of radiation to the tumor tissue without damaging the normal tissue surrounding the tumor [107].

IONPs have been extensively studied as radiosensitizers to decrease the radiation dose to normal tissues while increasing the dose to tumor tissues [50,78]. Radiation therapy increases the mitochondrial production of superoxide anions, which convert superoxide dismutase to hydrogen peroxide. Since IONPs can catalyze the conversion of hydrogen peroxide to highly reactive hydroxyl radicals, more reactive oxygen species (ROS) can be generated in tumor cells, thereby enhancing the efficacy of radiotherapy. In addition, surface−functionalized IONPs can enhance the radiosensitization effect and transport drugs to improve the antitumor effect [108]. Forrest et al. developed an IONP−based nanocarrier that could protect the efficient delivery of anti−Ape1 siRNA into brain cancer cells, enabling brain cancer to be sensitive to radiotherapy by knocking down the multifunctional DNA repair enzyme apurinuclease 1 (Ape1) and, thereby, increasing the antitumor effect and extending the survival of animals, synergizing with radiotherapy [109]. In a separate study, the researchers coated SPIONs with biocompatible, biodegradable coatings of chitosan, PEG, and PEI to achieve the targeted delivery of anti−Ape1 siRNA to pediatric brain tumor cells, reducing Ape1 expression by more than 75% and Ape1 activity by 80% in medulloblastoma and ependymoma cells [110].

### 3.6. IONPs for Combined Magnetic Hyperthermia Therapy (MHT)

IONP−mediated MHT is a recently proposed cancer treatment [111]. In this strategy, IONPs are applied to generate overheat via Brownian and Neelian relaxations with the assistance of an alternating magnetic field (100–300 kHz), thereby inducing tumor cell apoptosis by heating them to an over−high temperature of 42–46 °C [50,98]. Several studies have applied IONP−mediated MHT with IONP−mediated gene therapy for synergistic effects. For examples, Jiang et al. developed a magnetic nucleic acid delivery system comprised of IONPs and cationic lipid−like materials. It could efficiently deliver DNA and siRNA into cells and provide magnetic−guided targeting potential, allowing the combination of gene therapy and MHT [112]. In addition, local heating under MHT exposure increases microvascular tumor permeability, perfusion, and tumor cell membrane permeability, which is crucial for enhancing drug diffusion, cellular absorption, and drug action [98,113].

## 4. Impacts and Optimization of IONPs for Efficient Cancer Gene Therapy

During the whole process of delivering genes to tumor cells through IONPs, several barriers reduce the delivery efficiency: (i) the stability of nanoparticles; (ii) the accumulation to tumor sites; (iii) the efficiency of intracellular transport; (iv) the gene endosomal escape and further expression; and (v) short blood circulation and quick clearance by the immune system, which is closely related to the therapeutic efficiency of gene therapy [114,115,116,117]. Hence, further improvements for IONPs to overcome these barriers are highly required. Several vital aspects that determine the efficiency and safety of using IONPs for cancer gene therapy are discussed in the following section.

### 4.1. The Stability

Bare IONPs are easily able to aggregate into micrometer−sized clusters in the physiological environment, due to functional group interaction, as well as magnetic, and van der Waals forces [118]. The high surface charge also leads to the agglomeration after the nanoparticles are refined to the nanoscale [82]. Therefore, further modifications of IONPs to ensure colloidal stability are necessary, which is vital for both the delivery efficiency and the safety issues. Surface modification of IONPs with the same electronic charge is one of the major strategies to stabilize IONPs. This strategy can lead to repulsion between nanoparticles due to the electrostatic force, thus benefiting the improvement of nanoparticles’ dispersibility in aqueous solution [115]. For example, several polymers, such as chitosan [119,120], PEI [121,122], and PEG [123], have been applied for improving the colloidal stability of IONPs through the enhancement of electrostatic repulsion between nanoparticles.

In addition, generating repulsive steric forces between nanoparticles through surface modification using some long−chain molecules is another strategy to augment the colloidal stability of IONPs. The long−chain polymers attached to the surface of IONPs can increase the absolute value of the electric double layer on the surface of the nanoparticles and enhance the repulsive force between the particles to produce steric protection [124]. For instance, surface modification of IONPs with a methoxy−terminated PEG chain (5000 Da) showed no sign of aggregation for four months in deionized water at room temperature [125]. Long−chain molecules coating IONPs provided nanoparticles with stability through the optimization of the steric properties, further enhancing the repulsion between nanoparticles [126].

The adsorption of biomacromolecules on the surface of IONPs is another challenge impacting the tumor−targeted gene delivery efficiency. For example, IONPs are usually modified to a positive charge in the surface for the efficient loading of nucleic acid drugs. However, these positively charged nanocomplexes are easily adsorbed with the negatively charged plasma proteins in blood circulation after administration, which is known as the formation of protein corona on nanoparticles [127,128,129,130]. The formation of protein corona is believed to negatively affect the circulation time and the delivery efficiency of payloads [131,132,133]. For example, the interaction between nanoparticles and plasma proteins may alter the uptake and clearance of nanoparticles, thereby affecting the delivery efficiency. It was observed that the protein corona of SPIONs increased the endothelial permeability and uptake of endothelial cells [134]. The increase in hydrodynamic size provoked by the formulation of the protein corona also was reported to drive uptake mechanisms, such as macropinocytosis, further influencing the uptake of nanoparticles [135]. Moreover, the adsorption of plasma proteins on nanoparticles may also induce preferential cellular uptake by immune cells through the recognition of specific complement proteins in the corona, thus decreasing the biodistribution of nanoparticles in tumor cells and resulting in a rapid clearance of nanoparticles [136]. In addition, the adsorption of plasma protein or other kinds of biomacromolecules on IONPs may decrease stability and induce the aggregation of IONPs into micron−sized clusters [137]. For example, Safi et al. observed that the formation of protein corona had significant impacts on the dispersibility of IONPs in biological fluids [138].

Therefore, adopting surface modification to reduce the formation of protein corona on the surface of IONPs is of great significance to reduce clearance by the immune system, increase the blood circulation time, and improve the colloidal stability in biological fluids. For instance, Groult et al. showed that surface modification using phosphatidylcholine could prolong the circulation time and enhance the stability of IONPs due to the resistance of the protein’s adsorption [139]. In addition, PEG with high molecular weight was reported to reduce protein adsorption and non−specific uptake by macrophage cells, assisting IONPs escaping from the reticular−endothelial system for a long blood circulation half−life [140].

### 4.2. The Toxicity Induced by Functional Modification

The toxicity of IONP carriers remains a significant concern, and the modification materials play an important role in the toxicity of IONP−based vectors [70,141]. For uncoated IONPs, the LD−50 ranges from 300 to 600 mg kg^−1^. However, when their surface was coated with carboxydextran, the LD−50 value changed to 35 mmol kg^−1^, while it increased to 2000–6000 mg kg^−1^ when coated with stable and biocompatible dextran molecules [142,143]. In recent years, combining iron oxide cores with different coating molecules has greatly improved gene drug delivery to tumor sites. However, due to their varying toxicities, degradation rates, and pharmacokinetic properties, the study of vector toxicity has become more complex and crucial.

It is common practice to improve gene transfection efficiency by grafting PEI, the “golden standard”, onto IONPs. However, due to its high positive charge density, PEI modification alone would damage the cell membrane and induce cytotoxicity [144]. Therefore, Kievit et al. constructed a PEI−PEG copolymer by grafting PEG to low−molecular−weight PEI. This copolymer provided a physical barrier between the cells and the PEI, thereby reducing the potential toxicity of IONPs. Furthermore, it is essential to note that the design of this PEI−PEG−grafted chitosan had little impact on the magnetism and relaxivity of the iron oxide core, thereby endowing the platform with the dual benefits of gene delivery and in vivo imaging [54].

### 4.3. The Targeting Ability

Rapidly growing tumors develop intricate vascular structures that provide sufficient nutrients and oxygen to support tumor growth, resulting in a unique, perforated endothelial structure surrounding the tumor. This structure is highly permeable to IONPs, and it can facilitate the passive targeting of IONPs to solid tumors through the EPR effect [145]. In addition, the nanocore size and shape of IONPs play a crucial role in transporting genes into tumor cells.

It is believed that the optimal diameter of nanoparticles for cancer treatment should be between 10 and 100 nm [146,147,148]. When the diameter of IONPs is less than 10 nm, it is simple for them to pass through the tight endothelial junction and be eliminated by the kidneys’ first−pass elimination [149]. While penetrating deeply into the perivascular region of the tumor, the retention time of these small-sized IONPs may be brief because they are easily pushed out of the tumor by hydraulic forces. IONPs with a diameter over 100 nm are readily isolated by the spleen and liver and quickly absorbed by the mononuclear phagocytosis system [149]. Notably, nanoparticles larger than 100 nm are predominantly trapped in the extracellular space, which is not conducive to gene targeting in cancer cells [150,151]. The effect of tumor cell uptake correlates closely with the efficacy of gene therapy. Compared to larger nanoparticles, nanoparticles with a diameter of less than 50 nm and smaller nanointerfaces that form strong interactions by binding to cluster receptors on the cell membrane tend to have a higher uptake efficiency [134,152].

The regulation of nanoparticle shape is also crucial for enhancing the targeting efficacy of carriers, and an increasing number of studies suggests that adjusting nanoparticle shape can improve tumor targeting efficiency [150,153,154,155]. In the past decades, spherical nanoparticle carriers were the predominant shape of anticancer drug carriers due to their advantages, such as ease of fabrication. However, several reports have shown the promise of non−spherical nanocarriers have shown increasing promise in anticancer drug delivery [156,157]. Since different aspect ratios can affect the diffusivity of nanomaterials through pores and porous media, Vikash et al. designed and developed nanospheres and nanorods with tunable shapes but the same surface coating and discovered that nanorods excelled at passing through porous media and targeting tumors in vivo (Figure 6) [158].

By extending the circulation time of gene−carrying IONPs, it is possible to increase nanoparticle accumulation at tumor sites. The particle shape significantly affects the hydrodynamic behavior, influencing the circulation time. High−aspect−ratio nanoparticles can conform to blood flow and reduce collisions with vessel walls, resulting in a longer half−life in circulation. In addition, such nanoparticles can increase the probability of particle attachment to vascular walls and improve the interaction between receptor−targeting ligands, thereby facilitating tumor targeting (Figure 7) [159].

Park Ji−Ho et al. designed a dextran−encapsulated iron oxide chain−like aggregate, also known as a nanoworm. The magnetic nanoparticles can be aligned with the assistance of high−molecular−weight dextran chains, taking advantages of high aspect ratio. They discovered that the elongated structure increased the nanoparticles’ circulation time, their ability to attach to tumors in vitro via multivalent interactions with cell surface receptors, and their passive accumulation at the tumor site [157]. Other studies have demonstrated that extending nanoparticles in a one−dimensional direction can help them avoid natural elimination and achieve a longer circulation time, which is crucial for optimizing the shape of IONPs to improve their efficacy [160,161].

Malignant cells exhibit altered gene and protein expressions, such as overexpression of G−protein−coupled receptors, growth factor receptors, interleukins, transferrin, and polysaccharides, frequently used as targets for the active targeted delivery of functionalized carriers, resulting in enhanced drug delivery at the tumor site [162,163]. It is difficult for bare IONPs to rely solely on the EPR effect to deliver nucleic acid drugs to tumor growth sites such as brain tumors. However, such a situation can be improved to a certain extent under an external magnetic field. Active targeting based on the precise recognition of targeting ligands and their cognate receptors provides additional opportunities for IONP−based drug delivery [164]. Commonly employed targeting ligands in modifying IONPs include peptides, antibodies, aptamers, and small molecules, such as biotin, folic acid, and carbohydrates. Table 2 summarizes current active targeting modification strategies based on IONPs for the delivery of drugs to various drug sites. The active targeting modification strategies can not only improve the efficiency of tumor targeting but it can also prevent damage to normal cells and tissues caused by off−target effects during drug delivery, thereby enhancing the safety of gene therapy.

### 4.4. Intelligent Drug Delivery Based on IONPs

In recent years, intelligent drug delivery systems tailored to the tumor microenvironment (TME), such as acidity, hypoxia, and enzyme imbalance, have provided a more efficient drug delivery strategy towards tumors. This system can transform the carrier’s invisible surface into tumor−targeting surfaces for intelligent drug delivery, achieving responsive drug release based on tumor and microenvironment characteristics [172,173,174].

It has been reported that tumor cells have a different energy metabolism pattern in comparison with normal cells. They acquire energy mainly through anaerobic glycolysis due to the Warburg effect and generate large amounts of lactate, ATP hydrolyzate hydrogen ions, and carbon dioxide in tumor sites, consequently leading to the acidification of TME with a lower pH value (6.5–6.8) than healthy tissues [175,176]. This acidic TME provides an important target for tumor−specific diagnosis and treatment.

Based on the weak acidity of TME, certain pH−sensitive materials have been applied as surface modifications on nanoparticles to develop intelligent drug carriers [175,177]. Cao et al. designed a novel pH−responsive surface−modified single−walled carbon nanotube (SWCNT) and synthesized SWCNT−PB (SPB) to deliver DOX and survivin siRNA synergistically. When the drug−loaded nanocarriers were internalized by cells and encountered acidic endosomes or lysosomes, DOX and survivin siRNA adsorbed on SWCNTs completed the drug release via a mechanism triggered by the protonation of SPB, diffusing from lysosomes to the middle of the cytoplasm. When loaded on SPB, siRNA and DOX could be released efficiently into the cytoplasm and nucleus of A549 cells and exhibited potent antitumor effects in vitro and in vivo (Figure 8) [178].

Together with IONPs, stimulation−independent cationic polymers, such as PEI and PEG, can efficiently achieve gene delivery in tumor cells. However, the rate of dissociation between genes and carriers is limited, and premature dissociation from carriers may result in acid hydrolysis or enzymatic degradation. In contrast, stimulus−dependent materials with high responses to biological stimuli can transform the physicochemical properties of the carrier from a tightly complexed state with genes to a decomposed state in response to stimuli provided in the endosome and cytoplasm, allowing the production of many structurally intact and highly active gene drugs. In addition, the gene and the carrier cannot be recombined, making initiating the gene therapy procedure challenging. For instance, siRNA must combine with RISC in a free state in the cytoplasm to initiate RNA interference. Therefore, the molecular affinity between the surface−modified responsive materials of IONPs and genes should be irreversible [179,180]. The transformation of genes and polymers from complexation to decomposition in response to tumor−site−specific stimuli is the rate−limiting step in intelligent gene delivery.

The toxicity of smart, responsive materials used for the surface modification of IONPs is an essential factor that cannot be overlooked. The toxicity of the carrier can be affected by the molecular weight, cation density, morphology, polymer−to−nucleic acid ratio, and biodegradability of the surface−modifying material. In order to improve the efficiency of gene delivery, for instance, specific surface modification strategies have a high molecular weight, cation density, or N/P ratio, which increases vector toxicity and non−specific gene silencing [181]. Therefore, optimizing the minimal in vivo and in vitro toxicity protocols for gene therapy based on the molecular properties of various coating materials while simultaneously improving gene−targeted tumor delivery is of utmost importance.

## 5. Conclusions and Outlook

The magnetic responsiveness, good biocompatibility, ease of functional modification, and controllable synthesis of IONPs have enabled the possibility of using IONPs as vehicles for the targeted delivery of a variety of nucleic acid drugs targeting tumor tissues. In addition, the versatility of IONPs has further allowed the combination therapy of gene therapy with some other treatments, showing synergistic effects.

Despite the promising potential of IONP−associated tumor therapy demonstrated in animal studies, the reported clinic trials of this strategy, to the best of our knowledge, are currently limited. IONPs have been widely studied as a contrast agent for magnetic resonance imaging in clinical trials, such as in the diagnosis of pancreatic cancer (NCT00920023), breast cancer (NCT03243435), and thyroid cancer (NCT01927887). IONPs have also been proved by the US Food and Drug Administration (FDA) for the iron replacement therapy of anemia [88]. These clinical applications of IONPs, in addition to the successful results demonstrated in animal models, may suggest the potential translation of IONP−associated tumor therapy to the clinical bench.

However, there are still some challenges for the application of IONPs to deliver genes to cancer cells. Bare IONPs are generally regarded as nanomaterials with good biocompatibility and safety, but their surface modification may bring the risk of cytotoxicity. In addition, surface modifications of IONPs have demonstrated the advantages of offering various functions, but the magnetic response of IONPs may be affected, thus influencing the magnet targeting efficacy [149,182]. Therefore, it is essential to have further research on the surface−modified materials of IONPs. Moreover, many reports have demonstrated the tumor targeting ability of IONPs, but few studies had focused on the retention time of IONPs in tumors sites. Further improvements in the retention time of IONPs in tumor sites is still worth noting. In addition, since the size and shape of IONPs have a significant impact on drug delivery efficiency, large−amount production of IONPs with uniform and controlled sizes and shapes is a challenge.

In recent years, stem cells have been applied for targeting delivery of gene-loaded IONPs, showing attractive advantages in more specific tumor targeting and biocompatibility, which provide a novel strategy using IONPs for tumor−targeted gene therapy [183]. With the continues advances in materials science, nanotechnology, biological science, and gene therapy, it is believed that the development of an IONP−based delivery platform can be continuously improved and endowed with additional capabilities in cancer treatment.

## Figures and Tables

**Figure 1 nanomaterials-12-03323-f001:**
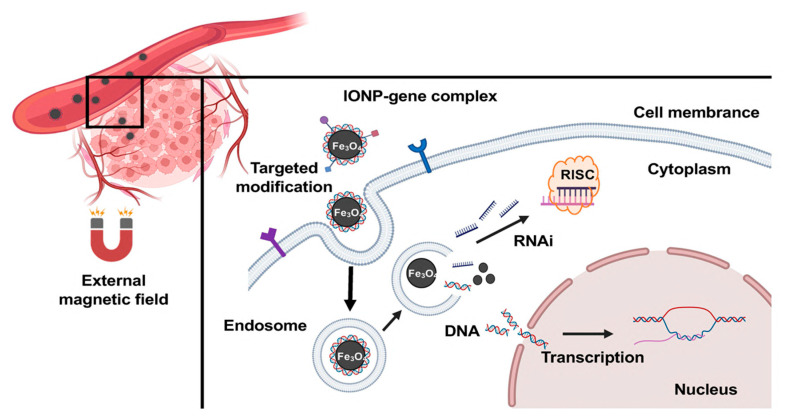
Schematic illustration of using IONPs to deliver gene drugs to tumor sites.

**Figure 2 nanomaterials-12-03323-f002:**
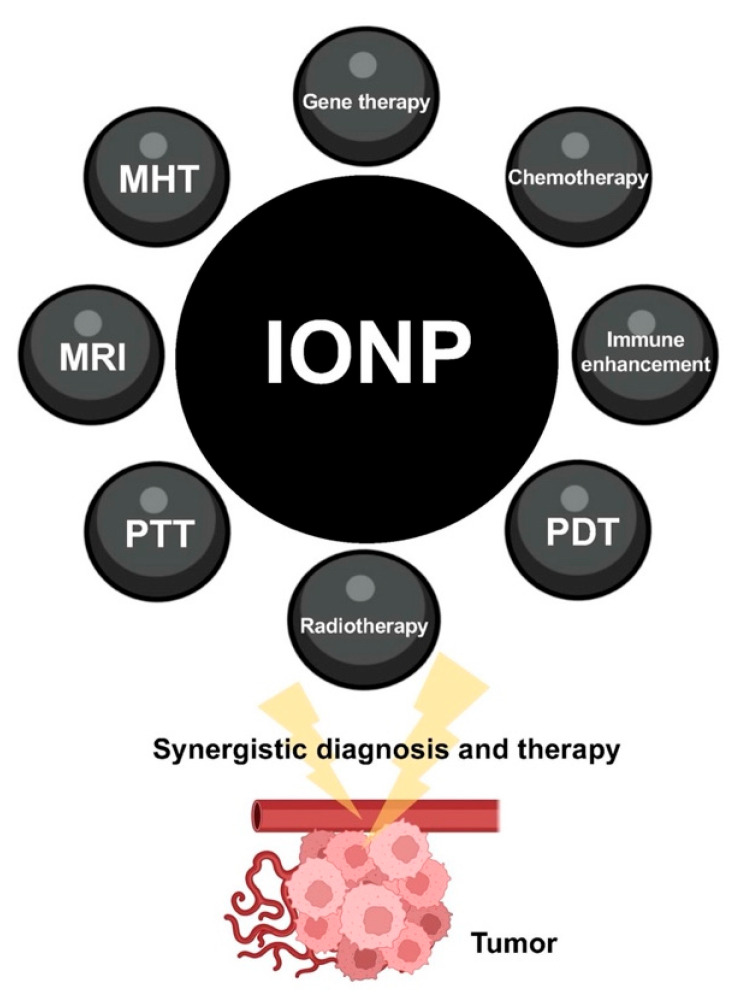
Gene therapy combining with other strategies to synergistically play a role in the diagnosis and treatment of tumors based on the versatility of IONPs.

**Figure 3 nanomaterials-12-03323-f003:**
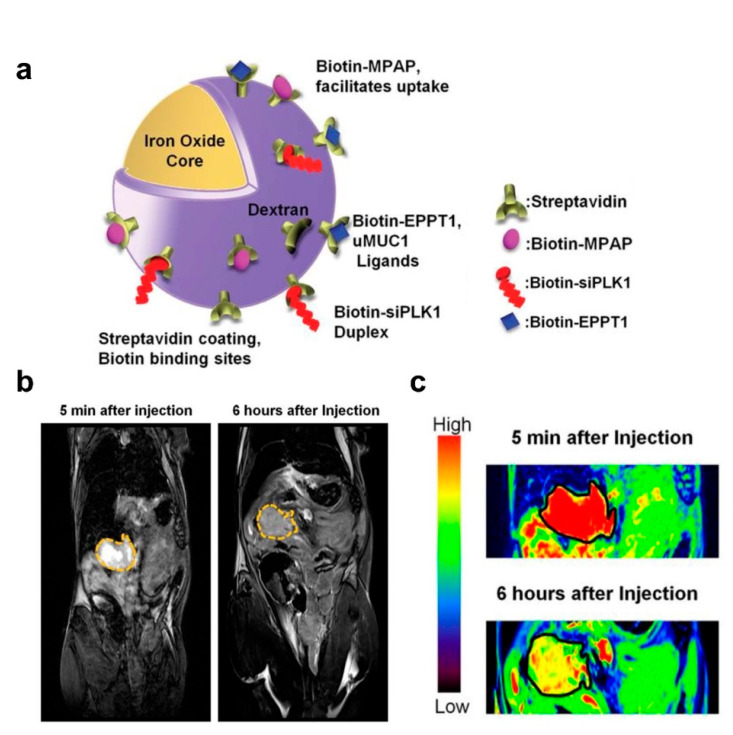
IONPs for gene delivery and in vivo tumor imaging. (**a**) Schematic representation of siPLK1−StAv−SPIONs. (**b**) In vivo MRI of mice bearing syngeneic orthotopic tumors was performed before and 6 h after intravenous injection of siPLK1−StAv−SPIONs. The dashed line marks the periphery of the tumor. (**c**) Color contrast images show decreased T2 relaxivity compared to pre−injection. Reprinted with permission from Ref. [86]. Copyright 2016, BMJ Publishing Group Ltd. and British Society of Gastroenterology.

**Figure 4 nanomaterials-12-03323-f004:**
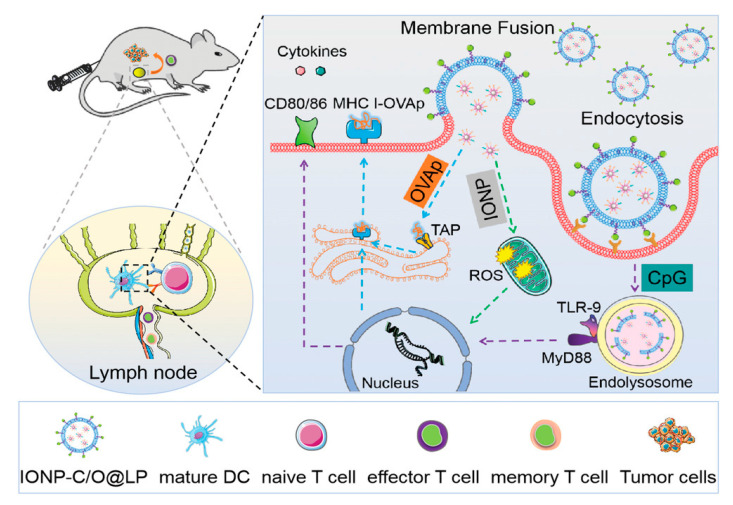
The synergistic effect and immune response elicited by IONP−C/O@LPs. IONPs co−delivered CpG DNA to active immature DCs, synergistically enhancing immune response and antitumor effect. Reprinted with permission from Ref. [97]. Copyright 2022, Wiley−VCH GmbH.

**Figure 5 nanomaterials-12-03323-f005:**
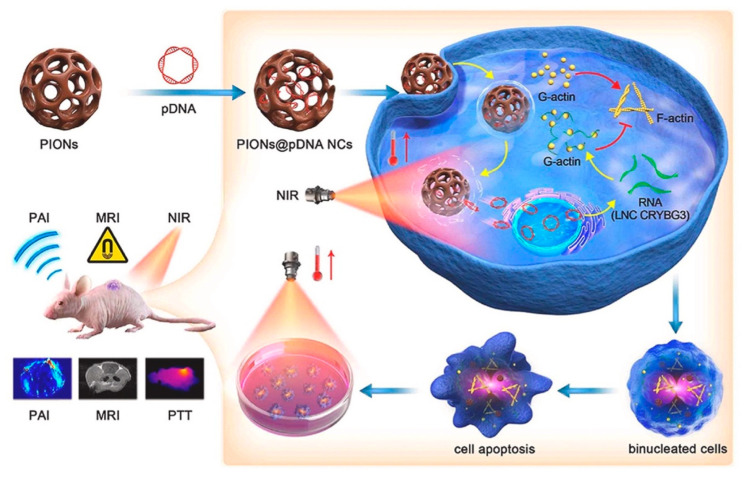
Porous iron oxide nanoparticles (PIONs) loaded with pcDNA3.1−LNC CRYBG3 nanocomplexes (PIONs@pDNA NCs) showed the synergistic ability of MRI, photothermal therapy, and gene therapy to achieve tumor−targeted therapeutics and diagnosis. Reprinted with permission from Ref. [103]. Copyright 2021, Elsevier.

**Figure 6 nanomaterials-12-03323-f006:**
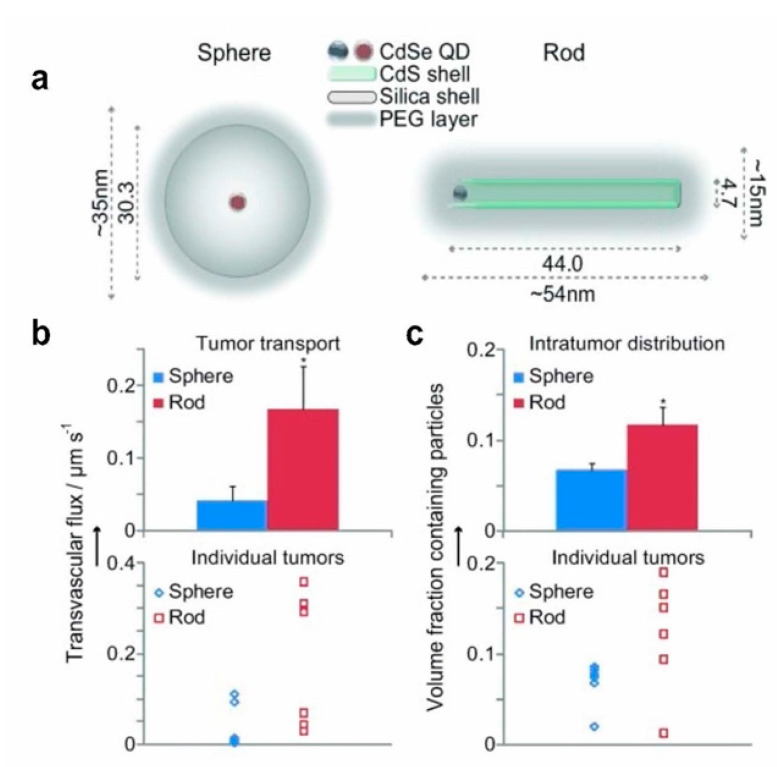
The effect of nanoparticle shape on tumor targeting. (**a**) Schematic diagram of nanospheres and nanorods. (**b**) Transvascular transport rates of orthotopic E0771 mammary tumors in mice. Nanorods were transported 4.1 times faster on container walls than nanospheres. (**c**) Nanoparticle distribution in mouse orthotopic E0771 mammary tumors. Nanorods penetrated 1.7 times the volume of distribution of nanospheres. Reprinted with permission from Ref. [158]. Copyright 2011, Wiley−VCH.

**Figure 7 nanomaterials-12-03323-f007:**
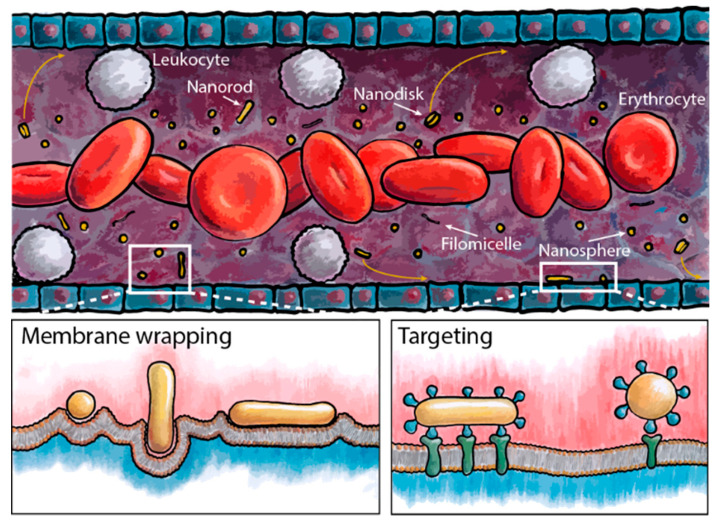
The size and shape of nanoparticles play a crucial role on the hydrodynamic behavior of particles in circulation, including the processes of membrane wrapping and targeting. Reprinted with permission from Ref. [159]. Copyright 2017, American Chemical Society.

**Figure 8 nanomaterials-12-03323-f008:**
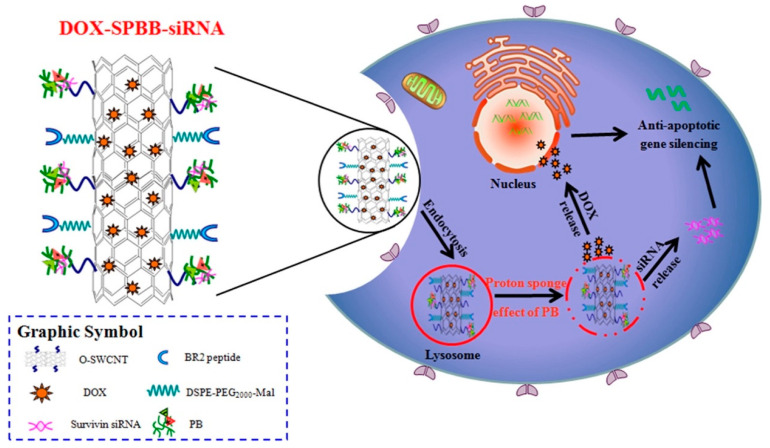
DOX−SPBB−siRNA nanocarriers release DOX and siRNA synergistically in A549 lung cancer cells according to the weak acidity of the tumor microenvironment. Reprinted with permission from Ref. [178]. Copyright 2019, American Chemical Society.

**Table 1 nanomaterials-12-03323-t001:** Examples of using IONPs for efficient gene delivery targeting to tumors.

Coating Materials	Size	Payload	Tumor Models	Efficiency	References
Chitosan, PEI (MW 3.9 kDA)	54.23 nm (core: 31.33 nm)	pDsRed−MAX−N1	4T1 breast cancer cells in vitro	High transfection efficiency	[53]
Chitosan, PEI, PEG	184 ± 6 nm (core: around 150 nm)	pEGFP−CS2	Xenografted tumor of C6 glioma	45.2 ± 3.4% transfected	[54]
Folic acid, lipo−polymersomes	220–260 nm (core: 170–220 nm)	pDNA	Xenografted tumor of Hela cervical cancer	High cellular uptake rate; high transfection efficiency	[55]
Fluorinated PEG−PEI	93.29 ± 7.31 nm	siRNA	4T1 breast cancer cells in vitro	More than 90% transfected	[56]
PEI	around 26.12 nm (core: around 7.95 nm)	siRNA	Ca9−22 oral cancer cells in vitro	BCL−2 mRNA level reduced to 18%	[57]
Calcium phosphate, PEG	67 ± 17 nm (core: 16± 3 nm)	siRNA	MDA−MB 231 breast cancer cells in vitro	VEGF mRNA level reduced to around 60%	[58]
PEG, PEI	79.2 ± 0.68 nm	siRNA	PC3 prostate cancer cells in vitro	Prostate cancer cell viability significantly decreased	[59]
Folic−acid−functionalized PEI	around 120 nm	siRNA	SGC−7901 gastric cancer cells in vitro	PD−L1 mRNA level reduced by 90.93 ± 0.79%	[60]
Tumor−targeting peptide, dextran	20–30 nm (core: around 20 nm)	miRNA−10b	MDA−MB−231 breast cancer cells in vitro	10b miRNA level reduced by 74%	[61]

**Table 2 nanomaterials-12-03323-t002:** Modification strategies for enhancing active tumor targeting of IONPs.

Modification Strategies	Tumor Models	Advantages	References
Transferrin	Orthotopic 4T1 breast cancer	Tumor retention levels 6 times higher than non-targeted nanoparticles	[165]
Wheat germ agglutinin	MDA-MB-231 breast cancer cells in vitro	Cancer cell death increased by about 2.5-fold	[166]
Folic acid cyclic Arg-Gly-Asp-D-Tyr-Lys	Orthotopic C6 glioma	Uptake enhancement through a combination of dual targets.	[167]
c(RGDyK), d-glucosamine	Xenografted tumor of 4T1 breast cancer	Tumor site accumulation and penetration depth increased	[168]
Monoclonal antibodies	Xenografted tumor of H460 lung cancer	In vivo ultrasound energy deposition significantly improved	[169]
PEGylated amphiphilic triblock copolymer	Xenografted tumor of U87MG glioma	Rapid clearance of the reticuloendothelial system avoided	[170]
Polyvinyl alcohol and Zn/Al-layered double hydroxide	HepG2 liver cancer cells in vitro	Antitumor ability increased No cytotoxic to 3T3 fibroblast cell	[171]

## Data Availability

Not applicable.

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
