# Peer review of "Biocompatible Iron Oxide Nanoparticles for Targeted Cancer Gene Therapy: A Review"

_nanomaterials, 2022, doi:10.3390/nano12193323_

Round 1
Reviewer 1 Report
The review by Zhang et al. concerns a very hot field in medical biotechnology (more than 20 reviews published in 2022 directly or indirectly concerning the subject). Authors’ aim is to review the latest progress in cancer gene therapy with IONPs. Theranostic synergies are underlined, as well as possible pitfalls and modification strategies to avoid these.
1) The most important limitation of this contribution is lack of focus on one or a few cancer types. Indeed, given the enormous and growing available literature, and the marked biological heterogeneity among the various forms of cancer, lack of focus leads to a “magic bullet” image of IONPs. Can the authors make some hypothesis on which forms of cancers should be more amenable to diagnosis(/treatment with IONPs?
2) Are there clinical trials in progress with IONPs? These should be discussed or, if they are lacking, this limitation should be cited and discussed, in the context of the numerous trials in progress with viral vectors.
3) Line 391. The pH values given are extracellular or intracellular? The sentence would seem to mean that “cells” are acidic (this is mostly an assumption) while the following sentence refers to “tumor microenvironment”. What are the consequences for IONPs effects in the two cases?
4) Line 282. “…by neutralizing the surface charges of nanoparticles”. Should not surface charge neutralization INCREASE aggregation? Please discuss.
5) Speaking of aggregation, the effect of adsorption of macromolecules present in biological fluids should also be discussed.
6) Lines 183-184. “Light-mediated therapy, including photodynamic therapy (PDT) and photothermal 183 therapy (PTT), plays an important role in the treatment and elimination of tumors….”. Can you provide some real-life examples in which this is true? If not, please specify “in experimental models”.
7) Language needs a really thorough revision. There are several typos, unclear sentences and uncommon expressions. Some examples:
a. Line 357. Figure 7 legend. “membrance” instead of “membrane”
b. Lines 170-172. The sentence is apparently incomplete (no verb in the principal sentence)
c. Line 122 “is capable” instead of “are capable”
Minor:
Tables 1 and 2 should indicate for all items also the tumor type not only the cell line involved. Moreover, in case of xenografted models, it should be indicated if the tumor was orthotopic or heterotopic.
Line 97. The correct EPR definition is “Enhanced Permeability and Retention” effect.
Reviewer 2 Report
Dear authors,
Please find the attached PDF document due to font size issues and alignment issues.

Round 2
Reviewer 1 Report
Authors have satisfactorily addressed my concerns.
Reviewer 2 Report
Accept in present form